# Self-Supervised Relationship Probing

**Jiuxiang Gu**[1], **Jason Kuen**[1], **Shafiq Joty**[2], **Jianfei Cai**[3],
**Vlad I. Morariu**[1], **Handong Zhao**[1], **Tong Sun**[1]
[1]Adobe Research, [2]Nanyang Technological University, [3]Monash University
{jigu,kuen,morariu,hazhao,tsun}@adobe.com,
srjoty@ntu.edu.sg, jianfei.cai@monash.edu

## Abstract

Structured representations of images that model visual relationships are beneficial for many vision and vision-language applications. However, current human-annotated visual relationship datasets suffer from the long-tailed predicate distribution problem which limits the potential of visual relationship models. In this work, we introduce a self-supervised method that implicitly learns the visual relationships without relying on any ground-truth visual relationship annotations. Our method relies on 1) intra- and inter-modality encodings to respectively model relationships within each modality separately and jointly, and 2) relationship probing, which seeks to discover the graph structure within each modality. By leveraging masked language modeling, contrastive learning, and dependency tree distances for self-supervision, our method learns better object features as well as implicit visual relationships. We verify the effectiveness of our proposed method on various vision-language tasks that benefit from improved visual relationship understanding.

## 1   Introduction

Visual relationships that describe object relationships in images have become more and more important for high-level computer vision (CV) tasks that need complex reasoning [1, 2, 3, 4]. They are often organized in a structured graph representation called scene graph, where nodes represent objects and edges represent relationships between objects. In recent years, we have witnessed great progress with visual relationship datasets such as Visual Genome [5] and the application of scene graphs to various CV reasoning tasks such as image captioning [6, 7], image retrieval [8], and visual reasoning [9].

Despite this, current visual relationship models still rely on human-annotated relationship labels. Due to the combinatorics involved — two objects and one relationship between them, where objects and relationships each have different types — relationships are numerous and have a long-tailed distribution and, thus, it is difficult to collect enough annotations to sufficiently represent important but less frequently observed relationships. Consequently, current visual relationship models tend to focus on modeling only a few relationships that have a large number of human annotations [10], and they ignore relationship categories with few annotations. We have seen some research attempts that use external knowledge databases to help enrich visual relationships, however, the total number of relationships modeled is still relatively small [11].

On the other hand, in the past few years, we have seen significant progress in natural language processing (NLP) towards building contextualized language models with self-supervised pretraining objectives [12, 13]. The removal of human annotators from the training loop has enabled training on massive unlabeled datasets, leading to significant advances in NLP performance [14, 15]. These trends have also brought significant advances in vision-language (VL) pretraining tasks [16, 17, 18, 19, 20]. Most existing VL pretraining methods concatenate visual objects and the corresponding sentences as one input and adopt the Transformer [21] as the core module to learn contextualized multi-modal representations in a self-supervised manner via self- and cross-attentions. These models rely heavily

on the multi-head attention layers to explore implicit relations, or they directly rely on attention distributions to explain the relations between objects [17, 22]. However, different layers vary in their behaviors [23, 24], and it has been shown that attention alone can be deceiving when used for interpretability and explanation [25]. Thus, existing VL pretraining algorithms suffer from two problems: discovered relationships are not modeled explicitly, but are instead expected to be implicitly represented as transformer weights; and, the concatenation of multimodal inputs at training time restricts the model to require multimodal inputs at prediction time, as well.

Motivated by textual relation mining work in NLP [26], we propose a novel framework that discovers dependencies between objects from the model's representation space which addresses the problems highlighted above. Our approach is based on two simple observations: *(1) when we slightly change the images, the relative visual relationships in those images remain unchanged; (2) relationships mentioned in image descriptions are visually observable in the corresponding image.* Our approach relies on three modules, each consisting of a set of layers. In the first module, implicit intra-modal relationships are modeled using transformer encoders. In the second module, cross-modal learning allows for implicit relationship information to be leveraged across modalities. In the third module, relationships between visual and linguistic entities are represented explicitly as latent variables via a technique we call *relationship probe*. All modules are trained using self-supervision, with a first stage relying on masked language modeling to train the first two modules, and a second stage relying on contrastive learning and linguistic dependency trees as supervisory signals to train the relationship probe network.

Our main contribution is a novel self-supervised relationship probing (SSRP) framework for finding dependencies in visual objects or textual entities that address issues with existing visual relationship models: it relies on self-supervision rather than explicit supervision, it explicitly models relationships as latent variables, and it leverages cross-modal learning but allows a single modality as input at prediction time. We conduct extensive experiments to demonstrate that our method can benefit both vision and VL understanding tasks.

## 2 Background

**Visual relationships.** It has been demonstrated that visual relationships between objects can help improve performance on many CV tasks [8, 27, 28, 29, 30, 31]. Most of these methods assume a known explicit graph structure, and limit the graph to the most frequently occurring predicate categories while ignoring others that do not have enough labeled examples. Relaxing this assumption, some works transfer the object representations learned with predicate functions to rare predicates in few-shot scene graph generation [32, 33, 34]. Other works capture the relations via attention mechanisms [35, 36, 37, 38]. However, unlike object detectors that are trained on unambiguous and objectively defined object class labels, visual relationships are subjective and it is hard to exhaustively annotate all possible relationships between objects. Thus, we do not explicitly define or label visual relationship classes, but instead, we discover the implicit visual relationships via the accompanied captions. We call our method SSRP in the sense that we do not use any explicit predicate labels.

**Pretraining.** Motivated by the huge success of BERT [13] in NLP, there is a growing interest in pretraining generic models to solve a variety of VL problems [39, 40, 22, 40, 18]. These methods generally employ BERT-like objectives to learn cross-modal representations from visual region features and word embeddings. They use self- and cross-attention mechanisms to learn joint representations that are appropriately contextualized in both modalities. However, most of the VL pretraining works heavily rely on massive amounts of visual-linguistic corpus [19, 17]. Moreover, although huge multi-modal training datasets enable pretraining methods to learn good representations for downstream multi-modal VL tasks, they usually do not benefit visual tasks that only deal with single visual modality during inference. We overcome this problem with a new approach that enables the generation of implicit visual object relationships even with only visual inputs during inference, while benefiting greatly from the cross-modality learning objectives during training.

We would like to point out that several works focus on investigating the representations learned by transformer-based pretraining models [41, 42]. Their findings suggest that BERT-based network pretraining learns a rich set of intermediate representations of both semantic and syntactic information, which can be used to unearth the representations of dependency grammar relations. An interesting finding in [26] shows that BERT can recover dependency parse trees that have not been encountered

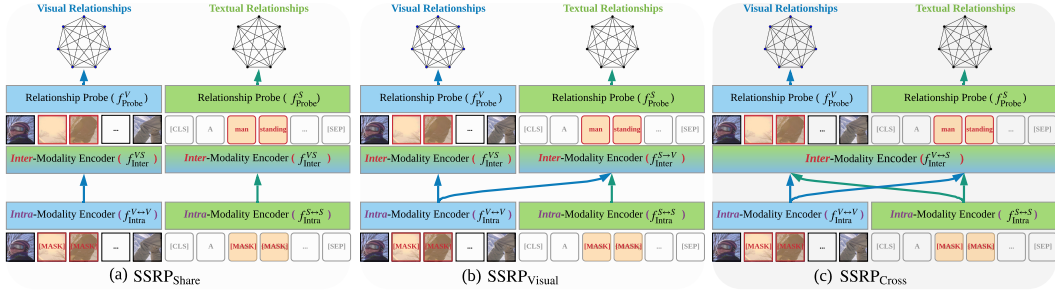

Figure 1: Overview of the proposed three types of SSRP frameworks, each of which consists of three types of modules: intra-modality encoder, inter-modality encoder and relationship probe.

during training. Coenen *et al.* [43] further present empirical descriptions of syntactic representations in BERT. These results in NLP motivate us to exploit BERT to find visual relationships between image regions without explicitly training on relationship annotations.

## 3 Method

Fig. 1 gives an overview of three variants of our method: $\text{SSRP}_{\text{Share}}$, $\text{SSRP}_{\text{Visual}}$ and $\text{SSRP}_{\text{Cross}}$. Each variant consists of three modules: intra-modality encoder, inter-modality encoder and relationship probe. The main difference among the three SSRP variants lies in the inter-modality encoding process. The intra-modality and inter-modality encoders are BERT-like encoders, that respectively capture implicit single-modality relations and cross-modality relations among the entities (image objects and textual tokens) and output contextual representations. The relationship probe generates relationship graphs for each modality from the encoded contextual representations in a self-supervised way.

In the following, we first briefly describe BERT [13] since our approach is based on BERT architecture, and then we describe the individual modules of our SSRP frameworks as well as the learning process.

### 3.1 Revisiting BERT

BERT uses Masked Language Modeling (MLM), a self-supervised pretraining objective that allows a transformer encoder [21] to encode a sequence from both directions simultaneously. Specifically, for an input sequence $S = \{w_1, \ldots, w_{N_w}\}$ of $N_w$ tokens, BERT first randomly masks out $15\%$ of the tokens and then predicts the masked tokens in the output. The masked tokens in the input sequence are represented by a special symbol [MASK] and fed into a multi-layer transformer encoder. Let $\boldsymbol{H}^l = \{\boldsymbol{h}_1, \ldots, \boldsymbol{h}_{N_w}\}$ be the encoded features at the $l$-th transformer layer, with $\boldsymbol{H}^0$ being the input layer. The features at the $(l+1)$-th layer are obtained by applying a transformer block defined as:

$$\boldsymbol{H}^{l+1} = \text{LN}\Big(\text{LN}\big(\boldsymbol{H}^l + f_{\text{Self-Att}}^l(\boldsymbol{H}^l)\big) + f_{\text{FF}}^l\big(\text{LN}(\boldsymbol{H}^l + f_{\text{Self-Att}}^l(\boldsymbol{H}^l))\big)\Big) \qquad (1)$$

where LN stands for layer normalization [44], $f_{\text{Self-Att}}^l(\cdot)$ is a multi-headed self-attention sub-layer, $f_{\text{FF}}(\cdot)$ is a feed-forward sub-layer composed of two fully-connected (FC) layers, wrapped in residual connection [45] with an LN as specified in Eq. 1. The token representations in the final layer are used to predict the masked tokens independently.

### 3.2 Model architecture

**Input embeddings.** The input to the three SSRP pretraining models includes both visual and textual elements, where the former is defined as regions-of-interest (RoIs) in an image and the latter is defined as the tokens in a caption. Specifically, given an image $I$, we use Faster-RCNN [46] to detect RoIs $\{v_1, \ldots, v_{N_v}\}$ and take the feature vector prior to the output layer of each RoI as the visual feature embedding. For a caption $S$, we insert the special tokens [CLS] and [SEP] before and after the sentence, and use the WordPiece tokenizer [47] to split it into tokens $\{w_1, \ldots, w_{N_w}\}$. Apart from token and visual feature embeddings, we also add positional encoding to represent tokens. In particular, for token $w_i$, its input representation $\tilde{\boldsymbol{w}}_i$ is the sum of its trainable token embedding, positional embedding (index in the sequence) and segment (image/text) embedding, followed by an LN layer. Each object $v_i$ is represented by its positional feature (normalized top-left and bottom-right

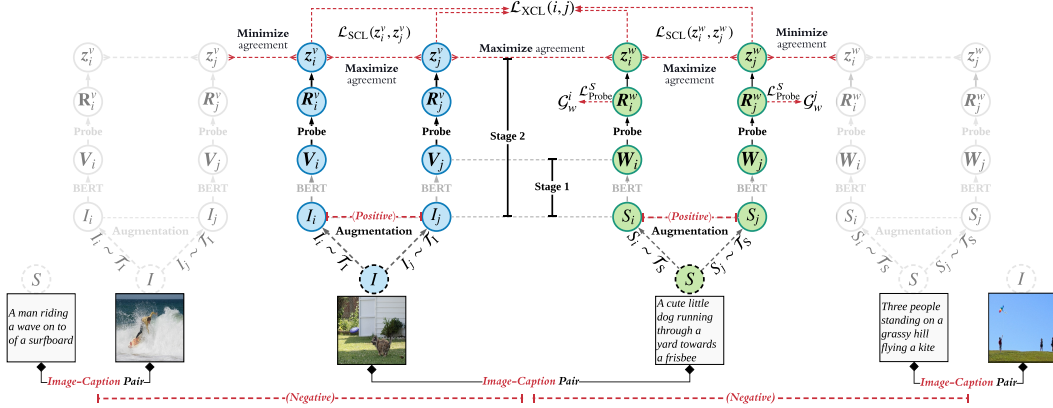

Figure 2: Illustration of our proposed learning process for relationship probing. The entire learning process consists of two training stages: training BERT encoders and training relationship probes. The notations $i$ and $j$ here refer to two different augmented images or sentences.

coordinates) and its 2048-dimensional RoI feature, both of which are transformed through FC+LN layers to obtain the position-aware object-level embedding $\tilde{\boldsymbol{v}}_i$.

**Intra-modality encoding.** The purpose of intra-modality encoding is to model the intra-relations of the encoded representations in one modality via self-attention, same as that in BERT. Specifically, we randomly mask out $\tilde{\boldsymbol{v}}_{\backslash i}$ and $\tilde{\boldsymbol{w}}_{\backslash j}$ with a fixed probability, and feed the masked object-level embeddings $\tilde{\boldsymbol{V}} = \{\tilde{\boldsymbol{v}}_1, \ldots, \tilde{\boldsymbol{v}}_{\backslash i}, \ldots, \tilde{\boldsymbol{v}}_{N_v}\}$ and word-level embeddings $\tilde{\boldsymbol{W}} = \{\tilde{\boldsymbol{w}}_1, \ldots, \tilde{\boldsymbol{w}}_{\backslash j}, \ldots, \tilde{\boldsymbol{w}}_{N_w}\}$ into two intra-modality encoders ($f_{\text{Intra}}^{V \leftrightarrow V}$ and $f_{\text{Intra}}^{S \leftrightarrow S}$) separately. Each layer in the intra-modality encoders contains a self-attention sub-layer and an FF sub-layer (Eq. 1).

**Inter-modality encoding.** The inter-modality encoder models the cross-modality relationships between image and textual entities. The three proposed SSRP pretraining models use different inter-modality encoding schemes as illustrated in Fig. 1. In SSRP$_{\text{Share}}$, the inter-modality encoding is done with a single encoder $f_{\text{Inter}}^{VS}$ that is shared between the two modalities, and $f_{\text{Inter}}^{VS}$ consists of a shared self-attention sub-layer wrapped in residual connection with an LN. The shared weights connect the two modalities by causing the projections of the two input modalities to align in the query, key, and value spaces. In SSRP$_{\text{Visual}}$, the textual features attend to visual features to connect the two modalities. In contrast to SSRP$_{\text{Share}}$, we keep $f_{\text{Inter}}^{VS}$ for the visual branch which contains a self-attention sub-layer and an FF sub-layer, while using $f_{\text{Inter}}^{S \to V}$ for the textual branch which consists of a self-attention sub-layer, one unidirectional cross-attention sub-layer, and an FF sub-layer. Finally, SSRP$_{\text{Cross}}$ uses an inter-modality bidirectional cross-attention encoder $f_{\text{Inter}}^{V \leftrightarrow S}$, where both textual and visual features attend to each other. Following [17], each layer in $f_{\text{Inter}}^{V \leftrightarrow S}$ consists of two self-attention sub-layers, one bi-directional cross-attention sub-layer, and two FF sub-layers.

**Relationship probing.** The purpose of the relationship probing is to model the implicit relations among visual or textual entities. Specifically, we build a latent relationship graph $\mathcal{G}_v$ for the objects in an image and a latent relationship graph $\mathcal{G}_w$ for the tokens in a caption, based on the unmasked contextual object representations $\boldsymbol{V} = \{\boldsymbol{v}_1, \ldots, \boldsymbol{v}_{N_v}\}$ and token representations $\boldsymbol{W} = \{\boldsymbol{w}_1, \ldots, \boldsymbol{w}_{N_w}\}$, which are the output feature vectors of the inter-modality encoders. Inspired by [26], we use a visual probe and a textual probe to compute the distances for each object pair $(\boldsymbol{v}_i, \boldsymbol{v}_j) \in \mathcal{G}_v$ and each token pair $(\boldsymbol{w}_i, \boldsymbol{w}_j) \in \mathcal{G}_w$, respectively. The distance for an object/token pair is defined as:

$$d_{\boldsymbol{B}_u}(\boldsymbol{u}_i, \boldsymbol{u}_j)^2 = (\boldsymbol{B}_u(\boldsymbol{u}_i - \boldsymbol{u}_j))^T (\boldsymbol{B}_u(\boldsymbol{u}_i - \boldsymbol{u}_j)) \qquad (2)$$

where $\boldsymbol{u} \in \{\boldsymbol{v}, \boldsymbol{w}\}$, $i$ and $j$ are the object/token indices, and $\boldsymbol{B}_u$ are the parameters for the probe layer. The learning goal of a structural probe (Sec. 3.3) is to determine the edge distances between all pairs of nodes. The outputs of the visual probe and the textual probe layer are respectively the distance matrices $\boldsymbol{R}^v = (d_{\boldsymbol{B}_v}(\boldsymbol{v}_i, \boldsymbol{v}_j)^2) \in \mathbb{R}^{N_v \times N_v}$ and $\boldsymbol{R}^w = (d_{\boldsymbol{B}_w}(\boldsymbol{w}_i, \boldsymbol{w}_j)^2) \in \mathbb{R}^{N_w \times N_w}$, which capture implicit relations between visual/textual entities.

### 3.3 Learning

We employ two learning stages in our method. In the first stage, we train the BERT encoders including the intra-modality encoders and the inter-modality encoders to obtain the contextual object representations $V$ and the token representations $W$. In the second stage, with these contextual representations, we freeze the BERT encoders and train the two probe layers to generate implicit relationship matrices $R^v$ and $R^w$. Fig. 2 shows a schematic diagram of our learning framework.

#### 3.3.1 Stage 1: Training BERT encoders

**Masked language modeling with RoI feature reconstruction.** We train the BERT encoders with the MLM objective to predict masked RoI feature $v_i$ and masked token $w_j$ given their surroundings $I_{\setminus i}$ and $S_{\setminus j}$. We also include a $L_1$ reconstruction smoothing loss [48] for the grounding of visual features. We minimize the following loss:

$$\mathcal{L}_{\text{MLM}} = -\mathbb{E}_{I,S \sim \mathcal{D}} \big[ \log p(v_i | I_{\setminus i}, \tilde{S}) + \log p(w_j | S_{\setminus j}, \tilde{I}) - \sum_i L_1(\boldsymbol{v}_i - g(v_i | I_{\setminus i}, \tilde{S})) \big] \quad (3)$$

where $\tilde{I}$ and $\tilde{S}$ are the image regions and input words with random masking, $g(.)$ outputs the unmasked visual feature, $p(v_i | I_{\setminus i}, \tilde{S})$ and $p(w_j | S_{\setminus j}, \tilde{I})$ are respectively the predicted probabilities for the target object label and word given the masked inputs, and $I$ and $S$ are sampled from the training set $\mathcal{D}$. Note that here we reuse the symbols $v$ and $w$ to represent both the visual features and the label/word for simplicity.

**Image-text matching.** An additional loss is added to perform the instance-level alignment between an image and its caption. Both positive ($y = 1$) and negative ($y = 0$) image-sentence pairs are sampled and the model learns to align with a binary cross-entropy loss:

$$\mathcal{L}_{\text{Match}} = -\mathbb{E}_{I,S \sim \mathcal{D}}[y \log p(\boldsymbol{f}_{\text{align}}) + (1 - y) \log(1 - p(\boldsymbol{f}_{\text{align}}))] \quad (4)$$

where $p(\boldsymbol{f}_{\text{align}})$ is the output probability of a binary classifier and $\boldsymbol{f}_{\text{align}}$ is the visual-textual alignment representation. For SSRP$_{\text{Share}}$ and SSRP$_{\text{Visual}}$, $\boldsymbol{f}_{\text{align}}$ is computed as $g_{\text{align}}([\bar{\boldsymbol{v}}; \boldsymbol{w}_{\text{CLS}}])$, where $\bar{\boldsymbol{v}} = \sum_i \boldsymbol{v}_i / N_v$ is the visual representation averaged over the contextual features of all the visual elements $V$, $\boldsymbol{w}_{\text{CLS}}$ is the contextual representation of the special token [CLS], and $g_{\text{align}}(\cdot)$ is a non-linear mapping function (see supplementary for details). For SSRP$_{\text{Cross}}$, we define $\boldsymbol{f}_{\text{align}} = g_{\text{align}}(\boldsymbol{w}_{\text{CLS}})$. Essentially, we force $\boldsymbol{w}_{\text{CLS}}$ to model either the aggregated textual or visual-textual information.

The overall training loss for the first-stage pretraining becomes: $\mathcal{L}_{\text{Stage1}} = \mathcal{L}_{\text{MLM}} + \mathcal{L}_{\text{Match}}$.

#### 3.3.2 Stage 2: Training relationship probes

In the second stage, the relationship probe layers are learned via a probe loss $\mathcal{L}_{\text{Probe}}^S$ and a contrastive loss $\mathcal{L}_{\text{CL-all}}$, where the former is to ensure the learned textual relationships $R^w$ is structurally consistent with a dependency tree and the latter is to ensure that the learned relationships $R^v$ and $R^w$ remain stable across different data augmentations.

In particular, on the language side, we use a pre-parsed dependency tree $\mathcal{G}_w$ for each sentence [49] to guide the textual relationship probe learning with $\mathcal{L}_{\text{Probe}}^S$ defined as:

$$\mathcal{L}_{\text{Probe}}^S = \frac{1}{N_w^2} \sum_{i,j} |d_{\mathcal{G}_w}(\boldsymbol{w}_i, \boldsymbol{w}_j) - d_{\boldsymbol{B}_w}(\boldsymbol{w}_i, \boldsymbol{w}_j)^2| \quad (5)$$

where $d_{\mathcal{G}_w}(\boldsymbol{w}_i, \boldsymbol{w}_j)$ is the distance between tokens $\boldsymbol{w}_i$ and $\boldsymbol{w}_j$ in the dependency tree $\mathcal{G}_w$.

For the contrastive loss, we adopt stochastic data augmentation methods to transform an original image (or sentence) into semantics-preserving data samples, and treat them as positive pairs; see Fig. 2, where $I_i \sim \mathcal{T}_I$ and $S_i \sim \mathcal{T}_S$ denote image and sentence augmentations, respectively.[1] For the data augmentation details, please refer to Sec. 4.1. Specifically, we sample a minibatch of $N_c$ image-caption pairs and apply two separate augmentation strategies to each modality, resulting in $2N_c$ image-caption pairs. For every positive pair, its negative pairs are not sampled explicitly, but

instead we take the other $2(N_c - 1)$ augmented image-caption pairs within a minibatch as negatives. We adapt the contrastive loss introduced in [50, 51] to our cross-modal scenario. The single-modality contrastive loss $\mathcal{L}_{\text{SCL}}(i, j)$ and cross-modality contrastive loss $\mathcal{L}_{\text{XCL}}(i, j)$ for a positive image-caption pair $\langle \{I_i, I_j\}, \{S_i, S_j\} \rangle$ are defined as:

$$\mathcal{L}_{\text{SCL}}(i, j) = -\log \frac{e^{\mathcal{Z}_{i,j}^{v,v}}}{\sum_{k=1}^{2N_c} 1_{[k \neq i]} e^{\mathcal{Z}_{i,k}^{v,v}}} - \log \frac{e^{\mathcal{Z}_{i,j}^{w,w}}}{\sum_{k=1}^{2N_c} 1_{[k \neq i]} e^{\mathcal{Z}_{i,k}^{w,w}}} \tag{6}$$

$$\mathcal{L}_{\text{XCL}}(i, j) = - \sum_{m \in \{i,j\}} \sum_{n \in \{i,j\}} \left( \log \Big( \frac{e^{\mathcal{Z}_{m,n}^{v,w}}}{\sum_{k=1}^{2N_c} 1_{[k \neq m]} e^{\mathcal{Z}_{m,k}^{v,w}}} \Big) + \log \Big( \frac{e^{\mathcal{Z}_{m,n}^{w,v}}}{\sum_{k=1}^{2N_c} 1_{[k \neq m]} e^{\mathcal{Z}_{m,k}^{w,v}}} \Big) \right) \tag{7}$$

where $1_{[k \neq i]} \in \{0, 1\}$ is an indicator function, $\mathcal{Z}_{i,j}^{x,y} = ((\boldsymbol{z}_i^{x\top} \boldsymbol{z}_j^y)/(\|\boldsymbol{z}_i^x\| \|\boldsymbol{z}_j^y\|))/\tau$ denotes the cosine similarity between $\boldsymbol{z}_i^x$ and $\boldsymbol{z}_j^y$, $\boldsymbol{z}^v$ and $\boldsymbol{z}^w$ are the nonlinear projections of vectorized relationship matrices $\boldsymbol{R}^v$ and $\boldsymbol{R}^w$ projected using MLP projection head [50], and $\tau$ is a temperature hyper-parameter [52]. The final loss is computed across all positive image-caption pairs in a mini-batch $\mathcal{L}_{\text{CL-all}} = \frac{1}{2N_c} \sum_{i,j} [\mathcal{L}_{\text{SCL}}(i, j) + \mathcal{L}_{\text{SCL}}(j, i) + \mathcal{L}_{\text{XCL}}(i, j)]$. Note that $\mathcal{L}_{\text{XCL}}$ is invariant to the order of sample indices $(i, j)$ and thus is included just once in $\mathcal{L}_{\text{CL-all}}$.

In this stage, the overall training objective is: $\mathcal{L}_{\text{Stage2}} = \mathcal{L}_{\text{Probe}}^S + \mathcal{L}_{\text{CL-all}}$.

## 4 Experiments

### 4.1 Datasets and implementation details

**Pretraining corpus.** To enlarge the training data, recent VL pretraining works [17, 16, 53, 18] use combined pretraining corpora such as Conceptual Captions (CC) [54], SBU captions [55], MSCOCO [56, 57, 58], Flickr30K [59], VQA [1], GQA [2], VG [5], BooksCorpus (BC) [60], and English Wikipedia (EW), *etc*. In contrast, we only aggregate pretraining data from the train (113k) and validation (5k) splits of MSCOCO [58]. Specifically, with each MSCOCO image associated with five independent caption annotations, MSCOCO provides us an aligned VL dataset of 591K image-and-sentence pairs on 118K distinct images. Table 1 summarizes the corpus used by different pretraining methods.

Table 1: Comparisons of the corpus used by different pretraining methods.

| Method | Source | Total | Method | Source | Total |
|---|---|---|---|---|---|
| LXMERT [17] | MSCOCO,GQA,VQA,VGQA,VG-Cap | 9.2M | VL-BERT [22] | CC,BC,EW | 3.3M |
| OSCAR [19] | MSCOCO,GQA,VQA,VGQA,CC,SBU,Flickr30K | 6.5M | VilBERT [16] | CC | 3.1M |
| UNITER [40] | MSCOCO,CC,VG,SBU | 5.6M | VisualBERT [53] | MSCOCO | 0.6M |
| Unicoder-VL [18] | CC,SBU | 3.8M | Ours: SSRP | MSCOCO | 0.6M |

**Data augmentation.** Instead of combining the existing VL datasets, we expand the pretraining corpus with data augmentation on both images and sentences, as shown in Table 2. For data augmentation on images, we employ horizontal flipping (HFlip) at the image level and a few augmentations at the RoI feature level including HFlip, rotations ($90^o$, $180^o$, and $270^o$) and bounding box jittering (with scale factors selected from the range of [0.8, 1.2]). We enrich the training sentences through two pretrained back-translators [61]: English→German→English (En-De-En) and English→Russian→English (En-Ru-En). Our augmentation strategies can generate significantly more training samples: 1.65M at RoI level and 1.77M at sentence level, while largely preserving the semantic information.

Table 2: Number of training samples at image, RoI, and sentence levels.

| Split | Image | | RoI features of Raw & HFlip images | | | Sentence | | |
|---|---|---|---|---|---|---|---|---|
| | Raw | HFlip | HFlip | Rotate(90°,180°,270°) | Jitter[0.8,1.2] | Raw | En-De-En | En-Ru-En |
| Train | 118k | 118k | 118k×2 | 354k×2 | 236k×2 | 591k | 591k | 591k |

**Pretraining setting.** We pretrain our three SSRP variants shown in Fig. 1. We set the numbers of layers for the intra-modality encoders of $f_{\text{Intra}}^{S \leftrightarrow S}$ and $f_{\text{Intra}}^{V \leftrightarrow V}$ to 9 and 5, respectively, and the number of layers for the inter-modality encoders of $f_{\text{Inter}}^{V S}$, $f_{\text{Inter}}^{S \rightarrow V}$, and $f_{\text{Inter}}^{V \leftrightarrow S}$ to 5. For each transformer block, we set its hidden size to 768 and the number of heads to 12. To keep the sizes the same for the relationship matrices, the maximum numbers of words and objects are equally set to 36.

Pretraining is divided into two stages. In stage 1, we train with $\mathcal{L}_{\text{Stage 1}}$. At each iteration, we randomly mask input words and RoIs with a probability of $0.15$. All models are initialized with BERT pretrained weights and the respective pretraining corpus is listed in Table 2. For cross-modality matching, we replace each sentence with a mismatched one with a probability of 0.5. We use Adam optimizer [62] with a linear learning-rate schedule [13] and a peak learning rate of $1e{-}4$. The training is carried out with four Tesla V100 GPUs with a batch size of 128 for 10 epochs. After stage 1, we freeze the parameters of the intra-modality and inter-modality encoders and further train the relationship probes with $\mathcal{L}_{\text{Stage 2}}$. The syntactic dependency tree for each sentence is built by [49]. All variants of SSRP are trained for 30 epochs with Adam, a batch size of 512, and a learning of $5e{-}5$.

**Fine-Tuning tasks.** We fine-tune the pretrained models to handle multiple downstream tasks: three VL understanding tasks (NLVR2 [63], VQA [1], and GQA [2]) and a generation task (image captioning), following the standard fine-tuning settings for downstream tasks in [17, 53]. For VL understanding tasks, we use linearly-fused probed relationships and visual-textual alignment prediction $f_{\text{align}}$ in Eq. 4 as features. For image captioning, we utilize the Up-Down [64] framework and incorporate the refined object features learned by SSRP$_{\text{Visual}}$. The captioning model is first trained with cross-entropy loss and is then followed by reinforcement learning loss [65].

## 4.2 Experimental results & analysis

We first perform ablation experiments over a few design choices of our method on NLVR2. We then show the comparison results on VQA, GQA and image captioning tasks.

**Effect of data augmentation.** Table 3 shows the ablation study results. For the 'Raw' setting, we pretrain our models only on the original corpus, while in the 'Aug.' setting, we augment the original corpus with the augmentation techniques mentioned in Table 2. It is evident that our data augmentation strategy indeed improves the performance of all three models. Note that we employ data augmentation only during pretraining, but not during fine-tuning.

Table 3: Ablation study on NLVR2. The reported results are accuracy numbers on Dev set.

| Method | $f_{\text{align}}$(Stage 1) | | $f_{\text{align}}$(Stage 1) + Rel.(Stage 2) | | |
|---|---|---|---|---|---|
| | Raw | Aug. | $R^v$ | $R^w$ | $R^v{+}R^w$ |
| SSRP$_{\text{Share}}$ | 60.53 | 61.67 | 62.52 | 62.66 | 64.25 |
| SSRP$_{\text{Visual}}$ | 69.92 | 70.75 | 71.23 | 71.24 | 72.03 |
| SSRP$_{\text{Cross}}$ | 74.35 | 74.48 | 74.25 | 74.68 | 75.71 |

**Effect of attention.** Comparing the three variants that use different attention settings in Table 3, we observe that SSRP$_{\text{Cross}}$ performs the best, and SSRP$_{\text{Visual}}$ is better than SSRP$_{\text{Share}}$. This confirms the benefits of the cross-attention structures that enable the features of one modality to attend to the other.

**Effect of relationship probing.** To analyze the effectiveness of the visual and textual relationships learned via pretraining, we concatenate the visual-textual alignment representation $f_{\text{align}}$ and relationships (Rel.) to form a relationship-aware feature vector for answer prediction. Table 3 shows that using language relationships $R^w$ leads to better results than using visual relationships $R^v$. This is due to the available dependency tree for supervising the language model during training, while the visual relationships are learned in a completely self-supervised way. Combining visual and textual relationships

Table 4: Online VQA/GQA results on the 'test-standard' splits, where '*' indicates the used corpus is larger than VisualBERT and ours.

| Method | VQA | | | | GQA |
|---|---|---|---|---|---|
| | Binary | Number | Other | Accu | Accu |
| BUTD* [64] | 86.6 | 48.6 | 61.5 | 70.3 | – |
| LXMERT* [17] | 88.2 | 54.2 | 63.1 | 72.5 | 60.3 |
| VilBERT* [16] | – | – | – | 70.9 | – |
| VL-BERT* [22] | 87.9 | 54.8 | 62.5 | 72.2 | – |
| OSCAR$_B$* [19] | – | – | – | 73.4 | 61.2 |
| UNITER-Base* [40] | – | – | – | 72.9 | – |
| VisualBERT [53] | 87.5 | 52.3 | 61.0 | 71.0 | – |
| SSRP$_{\text{Cross}}$ | 87.8 | 54.4 | 62.7 | 72.2 | 60.0 |

achieves the best results. Our method SSRP$_{\text{Cross}}$ (75.71) outperforms LXMERT (74.9) and Visual-BERT (67.4) on NLVR2 dev-set, demonstrating that the probed relationships are beneficial for the reasoning task.

**Results on VQA & GQA.** Table 4 shows the performance of our SSRP$_{\text{Cross}}$ on VQA and GQA. Our method outperforms VilBERT and VisualBERT, while being highly competitive with the best method that is trained with considerably larger training corpora.

**Results on image captioning.** Unlike the recent VL pretraining methods, which cannot be applied to single-modality vision tasks such as image captioning due to the cross attention used in pretraining, our SSRP$_{\text{Share}}$ and SSRP$_{\text{Visual}}$ models do not have such a limitation. Thus, we apply the stronger model SSRP$_{\text{Visual}}$ to image captioning using its refined object features and the learned implicit visual

relationships. Table 5 shows the quantitative results, where SSRP$_{\text{Visual}}$ outperforms the baselines, indicating that the learned relationship-aware image representations can benefit image captioning. Note that the online results of BUTD are achieved with model ensemble, while we use a single model.

Table 5: Results of image captioning on MSCOCO test split and online test server, where B@n, M, C and S are abbreviations for BLEU-n, METEOR, CIDEr, and SPICE, respectively.

| Method | B@1 | B@4 | M | C | S | Method | B@1 | B@4 | M | C | S |
|---|---|---|---|---|---|---|---|---|---|---|---|
| SCST [65] | – | 33.3 | 26.3 | 111.4 | – | Up-Down [64] (Our Impl.) | 81.2 | 36.9 | 28.3 | 120.8 | 21.6 |
| BUTD [64] | 79.8 | 36.3 | 27.7 | 120.1 | 21.4 | SSRP$_{\text{Visual}}$ | 82.0 | 38.1 | 28.8 | 126.7 | 22.3 |
| *Results on the online MSCOCO test server* | | | | | | | | | | | |
| BUTD [64] (c5) | 80.2 | 36.9 | 27.6 | 117.9 | – | SSRP$_{\text{Visual}}$ (c5) | 81.5 | 37.5 | 28.3 | 119.8 | – |
| BUTD [64] (c40) | 95.2 | 68.5 | 36.7 | 120.5 | – | SSRP$_{\text{Visual}}$ (c40) | 95.3 | 68.6 | 37.2 | 122.4 | – |

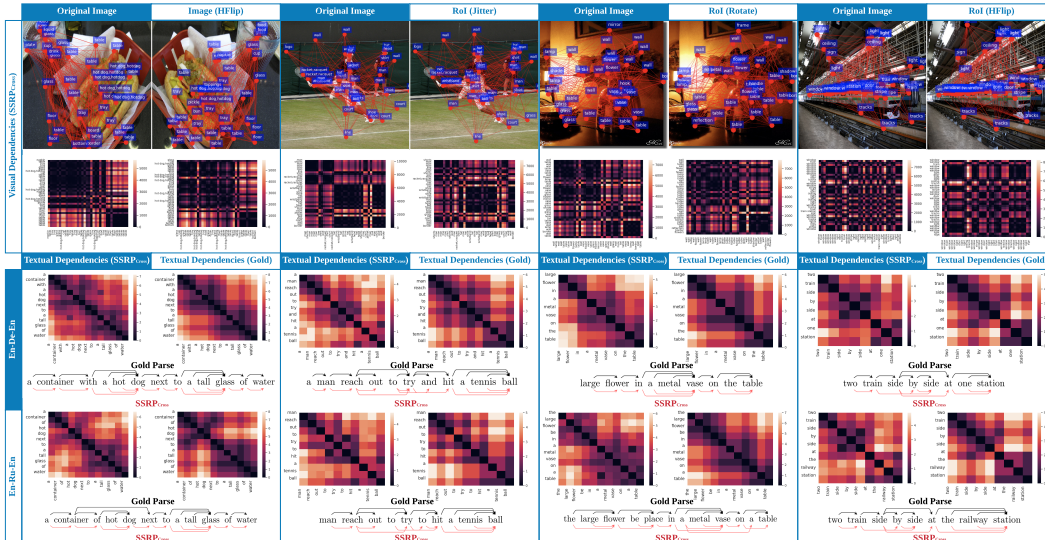

Figure 3: Examples of generated relationships for different augmented images and sentences. The bottom part shows the dependency trees resulted from SSRP$_{\text{Cross}}$ outputs. Black edges above each sentence are the gold tree provided by Stanza [49], and red edges are provided by our SSRP$_{\text{Cross}}$.

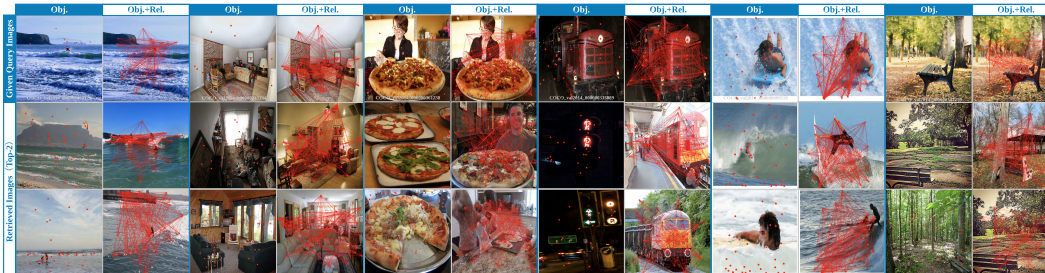

Figure 4: A visualization of the retrieved images on MSCOCO validation set. The 'Obj.' method averages object features and computes the cosine similarities between images. The 'Obj. + Rel.' method enhances the object features according to the predicted relationships.

**What do probes learn during training?** To answer that, we visualize in Fig. 3 the heat-maps of a few relationship examples generated by SSRP$_{\text{Cross}}$, where a darker color indicates a closer relationship. Particularly, the first row shows the example images and their augmented counterparts, each of which contains objects and their probed visual relationships represented by straight lines with varying color intensity values. The second row presents the visual relationship distance graphs for the corresponding images. The bottom rows show the distance graphs and dependency trees for augmented captions. Fig. 3 shows that the probed dependency trees closely resemble the gold dependency trees. In addition, the distance graphs of the original data samples and their augmented counterparts for sentences and images are also close to each other, validating our assumption that the visual/linguistic relationships should be preserved even when data augmentation is applied. Remarkably, the learned implicit

relationships between objects are stable across differently augmented images, despite the fact that no gold visual relationships are provided in training.

**Are visual relationships useful for visual tasks?** To further verify the benefits of implicit visual relationships in single-modality visual tasks, we perform image retrieval on MSCOCO with SSRP$_{\text{Visual}}$. Fig. 4 shows the top-2 image retrieval results. As shown, 'Obj. + Rel.' retrieves better visually-matching images that are consistent with the object relationships in query images. For example, in the third example, the person in the top-1 retrieved image is next to a pizza, similar to the original image. This suggests that our model can capture the complex underlying visual relationships.

## 5 Conclusion

We have proposed a self-supervised visual relationship probing method that implicitly learns visual relationships without training on ground-truth relationship annotations. Our method transfers the textual relationships from image descriptions to image objects and explores the visual relationships by maximizing the agreement between differently augmented images via contrastive learning. Through our relationship probes, we have demonstrated that relationship structures in images and sentences can be well explored with well-designed distance and contrastive learning objectives. We believe such implicit relationships in images and languages can help improve many existing vision-language tasks, especially in the scenarios with limited annotations.

## Broader Impact

Current representation learning models such as BERT and alike follow a similar structure. We think it is important to discover or probe the implicit knowledge that these models capture about language and vision. Our research on self-supervised relationship probing is a push in that direction and can be used for grounding the relationships expressed in language.

In this paper, we introduce SSRP, a self-supervised relationship probing method for visual and textual relationship extraction. Our research could be used to enrich the current scene graph generation methods and to complete the missing relationships between objects. The visual relationships generated by our method could be applied to a wide range of vision and vision-language applications including image captioning, image retrieval, object detection, visual question answering, visual reasoning, and visual-textual cross-modal retrieval, *etc*.

Here, we discuss the broader impact on the two important example applications (image retrieval and image captioning) which can benefit greatly from the implicit relationships obtained with our method. By performing image retrieval using the implicit visual relationships discovered with our method, visual search engines can provide higher-quality results that better respect the visual relationships contained in query images to users. This provides a smoother visual search experience and helps users find their desired images. On the other hand, for image captions/descriptions, with the implicit visual relationships generated by our method, richer and improved descriptions of images that more accurately describe the scenes in images can be obtained. This can help blind or visually-impaired people [66] 'see' their surrounding environments better.

In terms of technical impacts, our method opens a new direction to better model visual object relationships, which is completely different from current visual relation models that heavily rely on human-annotated explicit visual relation labels. Annotating visual relationships is a highly subjective process where different annotators are likely to annotate quite differently. Relations are also very diverse and there is no clear definition. Our approach bypasses all these challenges of annotating relations by advocating to discover rich implicit relations directly from natural images and their textual descriptions in a self-supervised manner without using any explicit relation annotations. Thus, our method leads to richer and fairer visual relation model.

In addition, in terms of dataset, our method also goes beyond current pretraining models that prefer to combine more and more datasets together for self-supervised training. Instead, our proposed method is developed specifically to work effectively with augmented data that can be cheaply obtained with the proposed augmentation strategies and can be nicely integrated into the self-supervision objectives.

Overall, our method makes VL pretraining and visual relationship modeling more accessible to the masses.

## Footnotes

[1]Note that in the interest of coherence, we describe data augmentation with contrastive learning in Stage 2, the augmented data can be used to train BERT encoders in Stage 1.

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
