[Supplementary Material]

# Appendix

## 0.1 Data augmentation

Fig. 1 shows some examples of augmented MSCOCO images and captions. We perform image-level augmentation to the input images of a Faster R-CNN (pretrained on the Visual Genome[1]) and apply RoI-level augmentation to the bounding boxes/RoIs detected by Faster R-CNN. For sentence augmentation, we use the transformer-based neural machine translation models [1] pretrained on WMT'19 [2] to perform back-translation. For ground-truth dependency trees, we parse each sentence with the dependency parser provided by Stanza[3].

Figure 1: Examples of augmented MSCOCO images and captions. For each augmented image, we show the object labels at the centers of respective bounding box for a better visualization. We apply per-category non-maximum suppression to the raw bounding boxes detected by Faster R-CNN.

## 0.2 Implementation details

### 0.2.1 Details of pretraining

Fig. 2 illustrates the visual-textual alignment mechanisms of the three variants of our proposed SSRP. For SSRP$_{\text{Cross}}$, we take the final hidden state of [CLS] to predict whether the sentence matches with the image semantically. For SSRP$_{\text{Share}}$ and SSRP$_{\text{Visual}}$, since they do not have the bidirectional cross-attention as in SSRP$_{\text{Cross}}$, we take $\sum_i \boldsymbol{v}_i / N_v$ as the additional input and concatenate it with $\boldsymbol{w}_{\text{CLS}}$ to generate the visual-textual alignment prediction using $g_{\text{align}}(\cdot)$.

Figure 2: An illustration of the mechanism used for obtaining the visual-textual alignment representation $\boldsymbol{f}_{\text{align}}$ for each of the three SSRP variants. These three SSRP variants can be used to facilitate the fine-tuning for different downstream tasks. Note that, SSRP$_{\text{Cross}}$ can only support visual-textual multi-modal downstream tasks such as VQA, while SSRP$_{\text{Share}}$ and SSRP$_{\text{Visual}}$ can support not only multi-modal downstream tasks but also single-modal visual tasks such as image captioning.

### 0.2.2 Details of NLVR2 fine-tuning

NLVR2 is a challenging visual reasoning task. It requires the model to determine whether the natural language statement $S$ is true about an image pair $\langle I_i, I_j \rangle$. During both fine-tuning and testing, we feed alignment representations of the two images and the probed relationships to a binary classifier. The predicted probability is computed as:

$$p(I_i, I_j, S) = \sigma\big(f_{\text{FC}}(f_{\text{p}}([\boldsymbol{q}_i; \boldsymbol{q}_j]))\big) \tag{A.1}$$

$$\boldsymbol{q}_k = f_{\text{q}}([\boldsymbol{f}_{\text{align}}^k; f_{\text{vw}}([\boldsymbol{R}_k^v; \boldsymbol{R}_k^w])]), \quad k \in [i, j] \tag{A.2}$$

where $\boldsymbol{f}_{\text{align}}^k$, $\boldsymbol{R}_k^v$, and $\boldsymbol{R}_k^w$ are the outputs of SSRP$(I_k, S)$, and $\sigma$ denotes the sigmoid activation function. The nonlinear transformation functions $f_{\text{vw}}$, $f_{\text{q}}$, $f_{\text{p}}$ and linear FC layer $f_{\text{FC}}$ have learnable weights.

For baseline models that do not consider relationships, the predicted probability is computed as:

$$p(I_i, I_j, S) = \sigma\big(f_{\text{FC}}(f_{\text{p}}([\boldsymbol{f}_{\text{align}}^i; \boldsymbol{f}_{\text{align}}^j]))\big) \tag{A.3}$$

We fine-tune all models (including SSRP) with sigmoid binary cross-entropy loss.

### 0.2.3 Details of VQA/GQA fine-tuning

VQA requires the model to answer a natural language question $Q$ related to an image $I$. We conduct experiments on the VQA v2.0 dataset. We fine-tune our model on the train split using sigmoid binary cross-entropy loss and evaluate it on the test-standard split. Note that VQA is based on the MSCOCO image corpus, but the questions have never been seen by the model during training. During fine-tuning, we feed the region features and given question into SSRP$_{\text{Cross}}$, and then output the alignment representation and the probed relationships that are fed to a classifier for answer prediction:

$$p(I, Q) = \sigma\big(f_{\text{FC}}(f_{\text{p}}(\boldsymbol{q}))\big) \tag{A.4}$$

$$\boldsymbol{q} = f_{\text{q}}([\boldsymbol{f}_{\text{align}}; f_{\text{vw}}([\boldsymbol{R}^v; \boldsymbol{R}^w])]) \tag{A.5}$$

where $\boldsymbol{f}_{\text{align}}$, $\boldsymbol{R}^v$, and $\boldsymbol{R}^w$ are the outputs of SSRP$_{\text{Cross}}$, and $\sigma$ denotes the sigmoid activation function. The nonlinear transformation functions $f_{\text{vw}}$, $f_{\text{q}}$, $f_{\text{p}}$ and linear FC layer $f_{\text{FC}}$ have learnable weights.

## 0.2.4 Details of image captioning

For image captioning, we use only the image branch of SSRP$_{\text{Visual}}$, and feed the unmasked image features into SSRP$_{\text{Visual}}$. For each input image, we first extract the contextualized visual representation $v_{1:N_v}$ and the implicit visual relationships $R^v$ from the pretrained SSRP$_{\text{Visual}}$. The inputs to the image captioning model are the refined object features $v_{1:N_v}$ and probed relationships $R^v$. We treat $R^v$ as a global representation for the image. We set the number of hidden units of each LSTM to 1000, the number of hidden units in the attention layer to 512. We first optimize the model on one Tesla V100 GPU using cross-entropy loss, with an initial learning rate of $5e-4$, a momentum parameter of 0.9, and a batch size of 100 for 40 epochs. After that, we further train the model to optimize it directly for CIDEr score [2] for another 100 epochs. During testing, we adopt beam search with a beam size of 5. We apply the same training and testing settings for Up-Down (Our Impl.) and SSRP$_{\text{Visual}}$.

## 0.2.5 Details of image retrieval

For image retrieval, we also feed the unmasked image features into SSRP$_{\text{Visual}}$ and obtain the refined contextualized visual representations along with the implicit visual relationships. We conduct the retrieval experiment on MSCOCO validation set. We randomly sample the query images and retrieve the top images according to their cosine similarities against the queries.

We compare two kinds of methods: one that uses contextualized visual representations $v_{1:N_v}$, and another one that uses both contextualized visual representations $v_{1:N_v}$ and implicit visual relationships $R^v$. For 'Obj. + Rel.' approach, we use the relationship-enhanced visual features obtained with $\frac{1}{N_v}\sum_i \frac{1}{N_v}\sum_k v_i d_{B_v}(v_i, v_k)^2$. For 'Obj.' approach, we simply average the contextualized object features with $\frac{1}{N_v}\sum_i v_i$. Fig. 3 shows the pipeline for the image retrieval task.

Figure 3: Illustrations of the two image retrieval methods mentioned in our paper.

 ## 0.3 Extra examples

Figure 4: Examples of generated relationships for different augmented images. Darker colors indicate closer visual relationships, while lighter colors indicate farther visual relationships.

Figure 5: Example of generated relationships for different augmented sentences. Bottom row shows the minimum spanning trees. Black edges are the ground-truth parse; red are predicted by $SSRP_{Cross}$.

## Footnotes

[1] https://github.com/airsplay/py-bottom-up-attention

[2] https://github.com/pytorch/fairseq

[3] https://github.com/stanfordnlp/stanza