[Reviews · NeurIPS 2020]

Review 1

Summary and Contributions: The authors developed a novel architecture for Visual Language tasks such as VQA, GQA and caption generation. For the first stage, They use Faster-RCNN to encode images and WordPiece tokenizer for the words. Both modalities uses BERT for learning good representations and a contrastive loss is used to match the two modalities. 3 different encoders are compared: i) a shared BERT encoder across both text and images ii) a BERT encoder that can attend the visual part when predicting text. iii) a cross modal BERT encoder that can attend both side. The second stage is the main contribution. they learn a relationship probe by computing pairwise distance between each embeddings for both modalities and making sure the resulting matrix are consistent across different data augmentation. Conventional data augmentation is used for images, while for text they used a pre-trained translator for going from En-De-En and En-Ru-En. For text, a supplementary supervised loss is used to aligned the relations with a pre-parsed dependency tree. Ablation study shows that the cross modality encoder provides much better accuracy on NLVR2. Also the relation probes provide a mild improvement over the cross-modal encoder. Other benchmarks shows consistent improvement over baselines.

Strengths: This work address an interesting challenge, which is the one of learning relationship between different parts of images and objects in an unsupervised fashion to target the long tail of rare relationships. The self supervised relationship probing is novel to my knowledge and is clever.

Weaknesses: A big weakness of the paper is the readability. The resulting algorithm has many parts, 2 training stage and total of 7 different component to the losses. that makes it really hard to follow and evaluate. It is also hard to relate to existing architecture e.g. I believe that the current architecture (beside relationship probing) is very similar to LXMERT [15] but it's not exactly clear what are the differences. Another weakness is experimental evaluation. While the ablation study is useful. It still doesn't highlight if it actually learns useful relationships. The gain could simply come from data augmentation provided by the pre-trained translator or the used parser. One of their contribution is to have competitive performance without the need of ancillary corpus by using the parser and translator. While it is interesting it makes comparison to baselines very difficult. Also, the qualitative evaluation of Figure 4 is highly unconvincing. ========= POST REBUTTAL ========== Rebuttal is well made and addresses my concerns. Table A Shows a good ablation study and Table B makes a clever comparison of blue score in their query matching. I've increased my score to 6.

Correctness: In both abstract and introduction, they justify this work by the long tail of relationships but none of this is really evaluated in the experiment setup. Ablation study highlight mild improvement when adding the relationship probe. This allows it to be better than the reported baseline but they omit to report UNITER, which has significantly higher accuracy. In table 5, they compare to BUTD which dates from 2017 on the leaderboard. I would assume that other algorithms came out since then?

Clarity: For a reader that is not savvy in the latest Visual-Language developments it is really hard to follow. I would recommend a major rewrite of the algorithm description with a more organised and concise approach. Also figures could be drawn differently to provide a more intelligible explanation of the algorithms e.g. the stack of layers is repeated 8 times in figure 2 and the horizontal lines showing the links between the positive and negatives are really confusing. Why not using 2 sub-figures, one that describes the stack of layers with more details and another that highlights how the different losses are composed. Also, there should be a figure for stage 1 and a figure for stage 2. Finally, the fact that v and w are used for both pre-bert embeddings and post-bert embeddings is extremely confusing. While we appreciate the many details for reproducibility purposes. I doubt I would be able to reproduce this work reliably mostly because of the confusion. There is no code included in the submission and no mention of open-sourcing the code. Also, I can't read Figure 3 with the maximum zoom of Mendely on my computer. I had to bring in another reader.

Relation to Prior Work: no. (see above)

Reproducibility: No

Additional Feedback:


Review 2

Summary and Contributions: The paper suggests a self-supervised approach to modeling relationships among entities in the same modality by leveraging cross-modal and intra-model relationships and evaluates the approach on a suite of vision-and-language tasks.

Strengths: Learning relationships without supervision, only by leveraging the intrinsic alignment between modalities, is a good idea. Scene graphs can be a powerful representation for many downstream tasks since they provide many useful abstractions, and doing away with the need for labels can bring obvious benefits in training them. Contrastive methods have seen much success in the self-supervised learning community and their application in learning relationship graphs is novel to my knowledge. The technique appears to lead to modest performance gains on NLVR2. The approach does not require massive amounts of data.

Weaknesses: While the method is referred to as self-supervised, it appears the authors use supervised methods to extract ground truth for object labels and sentence dependency trees. In particular, the dependency trees are used to guide learning of the relationship-graph-inducing distance metric in both vision and language. I am concerned as to whether this method relies on the strong structural information provided by GT parses to guide learning. The authors clarify they call their approach self-supervised w.r.t the lack of predicate labels, but there is still a good amount of supervision seemingly used. While there are some performance gains on NLVR2 as shown in Table 3, more ablations of the benefit of adding "Stage 2" would help further show the effectiveness of the method. In particular, since the approach is framed as one that is more so useful for downstream tasks than on its own, this evaluation seems a little weak. On the other hand, there is also not much analysis presented of the representation graphs that the model does learn. Both these factors combine to make it hard to judge the method's effectiveness as a whole. As an side, it is not clear to me how much variation is acquired by back-translation, or whether that variation is sensible. It may have been interesting to explore other methods of text augmentation.

Correctness: The methodology and approach seem sensible and correct, and the authors appear to follow common fine-tuning practices for evaluation. I am not sure whether the claim that the method is self-supervised can fully stand.

Clarity: Overall, I did find the paper well written and clear, but I had a few smaller remarks. 3.3.1, Eq 3 - How do you obtain p(v_i), p(w_j)? Presumably, p(w_j) is dot-product softmax over the whole text vocabulary? Is p(v_i) predicting the object class label as output by an off-the-shelf detector? If so, I believe it should be made clearer you do use supervision in the form of bounding box labels. Is the function g simply notation for accessing the hidden vector output by the intra-modality vision encoder at masked locations, or is there a learnable layer there? The phrase "outputs the unmasked visual feature" was not clear, since it seems like it outputs the ground truth RoI feature, but that doesn't seem to be the case. In the cases where image and text inputs are not aligned (for image-text matching), you still seem to use the representations output by the modality-combining encoder. Do you train on the MLM loss in these cases too (forcing the model to overcome misleading context)? Are the intra-modality encoders trained from scratch on the augmented MSCOCO or is the text encoder initialized with BERT pretraining? Typo on Line 191: Textural -> textual

Relation to Prior Work: Yes, the discussion compared to prior work seems complete. Whereas there are many difficulties in extracting relationships post-hoc from attention-based models, this paper suggests explicitly learning them.

Reproducibility: Yes

Additional Feedback:


Review 3

Summary and Contributions: The authors propose a self-supervised learning method that implicitly learns the visual relationships without relying on visual relationship annotations. The proposed method integrates several methods for self-supervision, and benefit various vision-language tasks.

Strengths: - The proposed self-supervised framework can learn visual relationships without using any relationship annotations, which avoids the limitations caused by manual labeling. - The experimental results show that the self-supervised learning method can benefit both vision and VL understanding tasks.

Weaknesses: - The proposed method is complicated, and it actually is the combination of a modified version of the masked language model and contrastive learning. So the contribution should be the application of these methods to implicit relationship learning but not a totally new framework. - Line 175, the authors say that both positive and negative image-sentence pairs are sampled. Since the image-text matching loss is applied at the same time with reconstruction loss in Stage 1, I think the authors should give a clearer explanation of how to sample single images and image pairs at the same time. - The SSRP method is complicated and contains various of losses or components for self-supervised learning. The author should provide more ablation results such as removing image-text matching loss. - In Table 4 and Table 5, the authors use different datasets or different settings compared to other methods. I am curious about what the performance is if training SSRP with the larger corpora like VL-BERT* in Table 4.

Correctness: Yes

Clarity: The paper is well-written.

Relation to Prior Work: Yes

Reproducibility: Yes

Additional Feedback:


Review 4

Summary and Contributions: The paper introduces a new and fresh idea compared to a lot of minor ablations that we are seeing in V&L pretraining domain. A self-supervised method is introduced which can implicitly learn the visual relationships in an image without relying on any ground truth visual relationship annotations thus breaking through the curse limited annotated visual relationship data which also has a long tail distribution problem. The method builds the intra- and inter-modality encodings separately and then uses relationship probing (contrastive learning loss and dependency tree) to discover the visual relationships between each modality. The method shows impressive results on multiple datasets and can be used by approaches which require vision-only embedding such as image captioning and improves these as well.

Strengths: -The method is novel and fresh and improves the existing object feature embeddings by building implicit visual relationship knowledge using self-supervised learning -Relationship probing further helps in improving the MLM pretraining representations. -The paper also introduces data augmentation techniques (though not novel) to gather more data for pretraining from the only source of COCO. -Results suggest that probing and the data augmentation both are useful. -Models intra- and inter-modality encodings separately which allows using the encodings in tasks which require inputs from only one modality. -Enhanced features help with image captioning task as well which improves the metrics compared to the original BUTD model when using them instead of original Faster RCNN ones. -The results on image retrieval and the annotated visual relationships are amazing given that they are trained in self-supervised way,

Weaknesses: I don’t have many concerns with this paper but I have some high-level issues that I believe should be addressed. - The effect of relationship probing hasn’t been studied independent of MLM training. Do we even need MLM or can be just get away with relationship probing. - The results on GQA are somewhat surprising compared to LXMERT. GQA is a task which should have better numbers with a better visual relationship understanding as the task depends on the scene graph itself. I understand that corpus for other datasets is larger, but can be know the number compared to VisualBERT or LXMERT only trained on COCO to clearly understand the actual impact. -To actually understand and for fair comparison, the number of parameters should also be compared between the different baselines and SSRP. - It would be good to have metrics on actual retrieval tasks or zero-shot caption retrieval to see how good the model is quantitatively along with qualitative results. -To understand the actual impact on the downstream tasks and the quality of the learned representations, it would make sense to test on low-resource tasks such as Hateful Memes dataset, OKVQA, TextVQA, TextCaps and nocaps for captioning. The current downstream task settings in the paper are data intensive and might not be capturing the full power of the model

Correctness: The methodology and claims seem correct. The numbers have been reported on online set.

Clarity: The paper was easy to read. I would suggest authors to be more descriptive with their captions. For example, the model figure is very hard to understand with current caption.

Relation to Prior Work: Yes, it mostly covers all of the literature. It might make sense to update the reference to include latest content. Some of the papers I mentioned and some of the latest content: 1. Agrawal, H., Desai, K., Wang, Y., Chen, X., Jain, R., Johnson, M., ... & Anderson, P. (2019). nocaps: novel object captioning at scale. In Proceedings of the IEEE International Conference on Computer Vision (pp. 8948-8957). 2. Singh, A., Goswami, V., & Parikh, D. (2020). Are we pretraining it right? Digging deeper into visio-linguistic pretraining. arXiv preprint arXiv:2004.08744. 3. Huang, Z., Zeng, Z., Liu, B., Fu, D., & Fu, J. (2020). Pixel-BERT: Aligning Image Pixels with Text by Deep Multi-Modal Transformers. arXiv preprint arXiv:2004.00849. 4. Kiela, D., Firooz, H., Mohan, A., Goswami, V., Singh, A., Ringshia, P., & Testuggine, D. (2020). The Hateful Memes Challenge: Detecting Hate Speech in Multimodal Memes. arXiv preprint arXiv:2005.04790. 5. Singh, A., Natarajan, V., Shah, M., Jiang, Y., Chen, X., Batra, D., ... & Rohrbach, M. (2019). Towards vqa models that can read. In Proceedings of the IEEE Conference on Computer Vision and Pattern Recognition (pp. 8317-8326). 6. Li, X., Yin, X., Li, C., Hu, X., Zhang, P., Zhang, L., ... & Choi, Y. (2020). Oscar: Object-semantics aligned pre-training for vision-language tasks. arXiv preprint arXiv:2004.06165. 7. Marino, K., Rastegari, M., Farhadi, A., & Mottaghi, R. (2019). Ok-vqa: A visual question answering benchmark requiring external knowledge. In Proceedings of the IEEE Conference on Computer Vision and Pattern Recognition (pp. 3195-3204).

Reproducibility: Yes

Additional Feedback: Update after rebuttal: I thank the authors for a very good rebuttal and their effort in addressing most of the reviewers concerns. I believe most of my concerns have been addressed and I would like to keep my score as it. The concern around GQA is still standing I would love to see if authors have any comments on it.

[Author Response · NeurIPS 2020]

We thank all the reviewers for their efforts and constructive comments! Most reviewers (R1, R2 and R4) think our method is novel, while having concerns on clarity and evaluation. Below we address the important and common issues.

**1. Supervision and ablation study (R1, R2, R3).** • Our method is self-supervised in the sense that it does not rely on any ground-truth visual relationship annotations, avoiding the challenging manual annotation of visual relationships; however, we do not restrict supervision for other learning components. • Our self-supervised relationship probing is based on the assumption that relations mentioned in image descriptions are visually observable. To model it effectively, we use dependency trees on the language side. • Although our framework consists of several modules and multiple losses, it is not complex at the conceptual level. Essentially, it contains two training stages, where the first stage resembles conventional BERT training, and the second stage is the proposed relationship probing. • In addition to Table 3, which shows the effectiveness of data augmentation and the proposed relation probing, we conduct additional ablation study to show the effect of the matching loss $\mathcal{L}_{\text{Match}}$ and the probe loss $\mathcal{L}_{\text{Probe}}$ in Table A. We can see that the matching loss is critical since it can help the model learn meaningful alignments between vision and language entities during training. On the other hand, the probing loss can further help improve the performance.

**2. Framework contribution (R1, R3).** • While $\text{SSRP}_{\text{Cross}}$ resembles LXMERT, we would like to empha-size that the goal of this paper is not to design a new BERT model for V&L. Instead, we learn visual relationships by self-supervised learning. Particularly, the designed relationship probe with its training process is novel and unique. As mentioned by R4, "this paper introduces a new and fresh idea compared to a lot of minor ablations that we are seeing in V&L pretraining domain". • Besides, our other two methods: $\text{SSRP}_{\text{Share}}$ and $\text{SSRP}_{\text{Visual}}$ are different from LXMERT. LXMERT and other V&L BERT-based models cannot be directly applied (without fine-tuning) to single-modality vision tasks such as image captioning due to the cross-attention used in pretraining, while our $\text{SSRP}_{\text{Share}}$ and $\text{SSRP}_{\text{Visual}}$ can.

Table A: Ablation study on NLVR2 Dev set. ✔/✗ indicates presence/absence.

| Method | Stage 1, Aug.(✗) | | Stage 1, Aug.(✔) | | Stage 1+2, Aug.(✔) | |
|---|---|---|---|---|---|---|
| | $\mathcal{L}_{\text{Match}}$(✗) | $\mathcal{L}_{\text{Match}}$(✔) | $\mathcal{L}_{\text{Match}}$(✗) | $\mathcal{L}_{\text{Match}}$(✔) | $\mathcal{L}_{\text{Probe}}^{S}$(✗) | $\mathcal{L}_{\text{Probe}}^{S}$(✔) |
| $\text{SSRP}_{\text{Share}}$ | 50.86 | 60.53 | 51.69 | 61.67 | 62.78 | 64.25 |
| $\text{SSRP}_{\text{Visual}}$ | 52.09 | 69.92 | 52.51 | 70.75 | 71.41 | 72.03 |
| $\text{SSRP}_{\text{Cross}}$ | 57.91 | 74.35 | 58.54 | 74.48 | 75.11 | 75.71 |

**3. Analysis of implicit graphs and Fig. 4 (R1, R2, R4).** The reviewers raised a good point that it would be better to provide quantitative results for image retrieval. However, it is hard to do so on MSCOCO since there are no officially annotated positive/negative pairs. As suggested by R4, we now report the results by calculating the sentence-level BLEU between captions (query image) and reference captions (retrieved images) in Table B. We see that 'Obj.+Rel.' outperforms 'Obj.', which shows the effectiveness of our implicit visual relationships for single-modality visual tasks.

**4. Language augmentation (R2).** We use two pivot languages, German (De) and Russian (Ru), and also different beam sizes to achieve diversity. As shown in supp., we can generate diverse captions and preserve the semantic meanings at the same time. We did explore other text augmentations, such as word substitution using BERT, *etc*. However, we found that they severely corrupt the original meanings. *E.g.*, given a sentence 'A large passenger airplane flying through the air', the back-translation provides 'A large passenger plane flying in the air', while the BERT-based word substitution gives 'a large passenger airplane flying through at night'.

Table B: Results on 1K query images randomly sampled from MSCOCO. We compute the BLEU scores based on all the associated captions from Top-$\{1, 5, 10\}$ retrieved images.

| Method | Top-1 | | Top-5 | | Top-10 | |
|---|---|---|---|---|---|---|
| | B@1 | B@4 | B@1 | B@4 | B@1 | B@4 |
| 'Obj.' | 38.28 | 6.11 | 45.37 | 6.18 | 48.46 | 6.28 |
| 'Obj.+Rel.' | 40.17 | 6.31 | 48.84 | 6.70 | 52.67 | 7.10 |

**5. Sampling strategies and larger corpora (R3).** For each training iteration, we sample a minibatch of image-caption pairs instead of individual images, as mentioned in Sec.3.3.2. The setting of this work is different from VL-BERT* trained on both visual-linguistic corpus and text-only corpus. This work is for discovering rich implicit visual relations directly from their textual descriptions. Thus, we can only use image-captions as pretraining corpus.

**6. Clarity and other issues. R1)** • The long-tail distribution of visual relationships mainly comes from the human annotations. Our models are not trained on such annotation. • We will improve the algorithm description and draw a simplified version for Fig. 2. • For the notations, we do not distinguish the input and output contextual features for simplicity. We will use different notations in the revised paper. • The main reason we compare with BUTD is that BUTD is a relatively well-tested framework, and also takes the object detection features as input similar to ours. Meanwhile, our best cider score (126.7) is close to that of SGAE[5] (129.1) which uses ground-truth visual relationships during training. During the submission period, UNITER[33] was not accepted to any venues, and thus we did not compare with it in Tab. 4. We will include it in our revised paper. • As for reproducibility, we will consider releasing the source code later. **R2)** • It is true that we use the object label predicted by the pre-trained object detectors, which can force the model infer the label by exploiting the linguistic clues when the corresponding RoI is masked out. $g(\cdot)$ is a non-linear mapping (learnable) layer (see the supplementary). • 'Outputs the unmasked visual feature' – it means predicting the unmasked features of the masked RoI input. • The text encoder is initialized with BERT pretrained model. • For unaligned image and text inputs, the MLM loss is still applied to each modality, but the alignment prediction training label is set to zero. **R4)** • We will consider low resource tasks and related papers in the revised version as per R4's suggestions. • The MLM loss helps to learn contextualized multi-modal representations in a self-supervised manner via self- and cross-attentions, our relationship probing generates relationship graphs in each modality from the encoded contextual representations. • The relative parameter size differences between the 3 variants are around $\pm 1\%$.

[Meta-Review · NeurIPS 2020]

Overall this is an interesting paper with very interesting ablations and experimental results. Authors also propose a new way of doing self-supervised learning. The results in this paper show that probing and the data augmentation both are useful. However, the writing of the paper requires some revision, at this stage the paper is a bit difficult to read which I think can be fixed with minor changes on the writing.